# Upcycling of Wood Dust from Particleboard Recycling as a Filler in Lignocellulosic Layered Composite Technology

**DOI:** 10.3390/ma16237352

**Published:** 2023-11-26

**Authors:** Anita Wronka, Grzegorz Kowaluk

**Affiliations:** Institute of Wood Science and Furniture, Warsaw University of Life Sciences—SGGW, Nowoursynowska St. 159, 02-776 Warsaw, Poland; anita_wronka@sggw.edu.pl

**Keywords:** upcycling, particleboard, plywood, dust, circular economy, filler

## Abstract

The following research aims to investigate selected properties of three-layer plywood, manufactured using dust from the milling of three-layer particleboard as a filler in the bonding mass. Four types of fillers were considered in the study: commercial rye flour, wood dust naturally occurring in the composition of particles used industrially for particleboard production, wood dust from the first batch of shredded particleboard, and dust from the second round of milled particleboard. The highest modulus of elasticity (MOE) values were observed for the reference samples. Notably, in the samples containing filler sourced from the secondary milling of particleboard, the MOE exhibited an upward trend in conjunction with increasing filler content. The modulus of rupture (MOR) decreased with an elevated degree of filler milling from 73.1 N mm^−2^ for the native filler, through to 68.9 N mm^−2^ for the filler after 1st milling, and to 54.5 N mm^−2^ for the filler after 2nd milling (with 10 parts per weight (pbw) of filler used as an reference), though it increased slightly as the filler content increased. The most favorable outcomes in shear strength were achieved in samples containing filler material from the initial milling of particleboard. The thickness swelling peaked in variants utilizing filler material from both the initial and secondary milling of particleboards (20.1% and 16.6% after 24 h of soaking for samples with 10 pbw filler after the 1st and 2nd milling, respectively, compared to 13.0% for the reference samples). Water absorption testing exhibited a more pronounced response in the newly introduced variants, although the samples containing filler from the initial and secondary milling processes eventually yielded results akin to the reference sample, with naturally occurring dust displaying higher water absorption values. The highest density values (about 1224 kg m^−3^) were observed in the reference samples. A similar density profile was recorded for samples with five parts of wood flour as filler, although the density of the bonding line was slightly lower in these instances (1130 kg m^−3^). This research confirms the feasibility of applying the aforementioned dust as an alternative to conventional fillers in plywood technology. It also raises the question of how to effectively remove glue residues from wood-based composite dust, which would enhance their absorption properties.

## 1. Introduction

Plywood is one of the most widely used wood materials and is employed in airplane construction, although nowadays it is not used so often in the furniture industry as 20 years ago [1]. It is also applied in the production of marine components, although lighter and more cost-effective alternatives are increasingly being sought [2].

In plywood technology, the components—not only veneers and adhesives, but also fillers—are increasingly being subjected to modifications. The aim of this research was to search for new potential applications of known raw materials that have not yet been fully examined or whose potential has not yet been fully exploited. Attempts have already been made to, among other things, modify the veneers through the utilization of environmentally friendly vacuum-assisted resin transfer molding (VARTM) technology, which has significantly increased the strength properties of plywood and reduced water absorption [3]. The impregnation of veneers can contribute to fire resistance properties if appropriate measures are employed [4]. Expanding societal awareness is forcing the woodworking industry to become increasingly environmentally friendly, which explains why various attempts have already been undertaken to replace formaldehyde-based thermosetting resins with adhesives of natural origin, e.g., glutaraldehyde-modified starch has been considered in this regard [5]. Studies have confirmed that glutaraldehyde-modified starch can be utilized as a binder substitute in plywood technology [6]. Other examples of plywood produced with no formaldehyde include plywood glued with poly(vinyl alcohol)-tannin-hexamine baize glue [7]. The plywood component modifications that have been investigated include the addition of a silica filler to the adhesive in order to reduce the penetration of selected chemicals. This type of plywood was developed for warfare purposes [8]. In turn, the wood bark used as a filler in plywood technology has shown properties that minimize free formaldehyde emissions [9,10,11]. Green tea also has the ability to decrease formaldehyde emissions, as confirmed in a study making use of green tea leaves as a filler in plywood technology [12]. Another raw material that enables the reduction of free formaldehyde emissions is a powder derived from modified ground pine needles. This filler allows the high strength parameters to be maintained, as well as contributing to the reduction of formaldehyde emissions. Tests were carried out considering urea-formaldehyde resin at a 1:10 weight ratio of modified needle powder to resin [13]. Research states that plants containing polyphenols, especially tannins, are characterized by their ability to reduce formaldehyde emissions [14]. Among such plants are thymus plants, which can be successfully applied as filler for plywood [15].

Rye flour is a commonly employed filler in plywood production due to its wide availability and low cost. Regarding the potential food crisis, it is beneficial to be aware of the alternative food raw materials that can be implemented interchangeably instead of rye flour, which is conventionally used [16]. Increasingly, researchers are also investigating new flours, as despite having a very similar texture, each new raw material can have slightly different properties. Such experimental attempts include chestnut flour [17], rice starch [18], biomass combustion fly ash [19], or soy flour [20,21,22]. These raw materials belong to the group of active fillers and can therefore swell, absorbing moisture. The best solution in this regard would be to replace the filler with dust, which is inedible. There are also known passive fillers, which do not react as intensively to moisture as active fillers. Additionally, a division of organic and inorganic fillers has been described [23]. This group includes fillers obtained from chestnut shell flour [24], eggshells [25], or reused coffee grounds [26]. In plywood technology, another alternative filler employed is derived from cactus waste seeds. These seeds have been successfully utilized as a filler in plywood panels bonded with phenol-formaldehyde adhesive. The decision to consider cactus waste seeds as a filler was encouraged by the substantially higher cellulose content they possess, which stands at 27%, in contrast to the previously employed olive seeds for this adhesive application. The integration of cactus seed filler led to notable enhancements in the strength properties of the plywood, yielding results that compare favorably with those of the benchmark plywood. Furthermore, an observed reduction in formaldehyde emissions was a noteworthy outcome associated with the use of cactus seed filler [27]. The use of fillers enables the viscosity of the glue to be controlled and prevents excessive absorption of the glue into the veneer, also allowing for the acceleration of dimensional stability [28,29]. Activated carbon, due to its structure, holds significant potential as a filler in wood-based material technology. It accelerates the curing of urea-formaldehyde (UF) resin and also has the ability to reduce formaldehyde emissions [30].

The increasing amount of waste is forcing society to search for new solutions and strategies for waste reuse. Another reason to expand research on this issue is the desire to follow the guidelines according to the principles of a circular economy, maximizing the use of raw materials before deciding to discard or dispose them. Following this statement, Russian scientists used Powdered Paper Resin Films (PRFs) consisting of melamine-urea-formaldehyde resin and bleached resin pulp for their research [31]. When combined with urea-formaldehyde resin, PRF increased the viscosity of the adhesive pulp up to 110%, simultaneously extending the gel time of the adhesive. This problem has been solved using the hardener MO-4CБ (Russian acronym). Ammonium chloride can also be applied for this purpose; however, its effect is weaker compared to MO-4CБ. The present adhesive mixture has been adapted for three-layer plywood, and the strength parameters with such a mixture increased by 5% on average. The next example of an environmentally friendly adhesive for plywood production is a binder created from polypropylene filters from single-use face masks [32]. The use of plastic containers as a bonding agent also aligns with the concept of a circular economy. Until now, such containers were properly cut and utilized as adhesives in plywood production. The research included the selection of pressing temperatures to ensure the best bonding quality and prevent degradation of the containers [33]. Another example of obtaining eco-friendly adhesives for plywood involves utilizing extracts derived from the bark of grey alder (*Alnus incana*) and black alder (*Alnus glutinosa*), both rich in condensed tannins (CTs). Combined with polyethyleneimine or ultra-low emitting formaldehyde resin, they achieve adhesive qualities comparable to industrial resin, while reducing formaldehyde emissions up to 60%. Each of the adhesive combinations produced met the requirements set by the EN 314-2:1993 [34] standards for both indoor and outdoor applications. This study confirms that alder bark particles can serve as a substitute for conventional fillers [35].

It is less challenging to find information on the use of the dust fraction generated during the production of wood-based composites [36] than on the use of dust generated from the recycling of wood-based composites, such as particleboard and plywood. An intriguing approach for the development of an environmentally friendly adhesive involves utilizing waste cottonseed protein and sawdust as key constituents. To enhance its adhesive properties, a dual crosslinking modification process was employed, incorporating Isophorone diisocyanate (IPDI) and oxidized cellulose (OC). The strength property results obtained from testing remain in compliance with the applicable Chinese strength standards for plywood, underscoring the considerable potential of this adhesive formulation as a sustainable alternative to urea-formaldehyde (UF) resin [37].

This research aims to explore the potential use of wood dust derived from recycled wood materials, such as particleboards, in an effort to identify new possibilities for their utilization before considering their disposal. In the scope of this research, fine fractions of recovered wood material with different contents were used as bonding mass fillers in plywood production. Then, the selected mechanical and physical properties of the manufactured plywood were evaluated to estimate the influence of the quality and quantity of the tested filler on the plywood properties (Figure 1).

## 2. Materials and Methods

### 2.1. Materials

This research involved the production of three-layer plywood, manufactured using birch veneers (*Betula* spp.). The veneers had a thickness of 1.8 mm, with a moisture content (MC) of about 6% and dimensions of 360 × 360 mm^2^.

The binder used was an industrial urea-formaldehyde (UF) resin Silekol S-123 (Silekol Sp. z o.o., Kędzierzyn, Koźle, Poland), containing about 66% dry content [38] with a molar ratio of about 0.9. Additionally, an ammonium nitrate water solution was used as a hardener, when subjected to a temperature of 100 °C reach the curing time about 86 s.

Rye starch was applied as a filler in the reference sample (producer: BioLife Sp. z o.o. ul. Miodowa 17, 17-100 Bielsk Podlaski, Poland).

The remaining fillers employed included wood dust derived from sifting conventional wood shavings utilized in the production of particleboard (native) and wood dust derived from grinding particleboard with a nominal density of 650 kg m^−3^. Then, particleboards with a density of 650 kg m^−3^ were produced again and ground. Dust was also obtained from this grinding, which served as the filler after 2nd milling. The size of the dust particles remained lower than 0.125 mm, and the bulk density equaled about 270 kg m^−3^. The entire procedure and characteristics have been described in [39]. Other studies have confirmed that processing a wood composite made from already fragmented wood contributes to the formation of smaller fractions compared to the processing of solid wood [40].

**Figure 1 materials-16-07352-f001:**
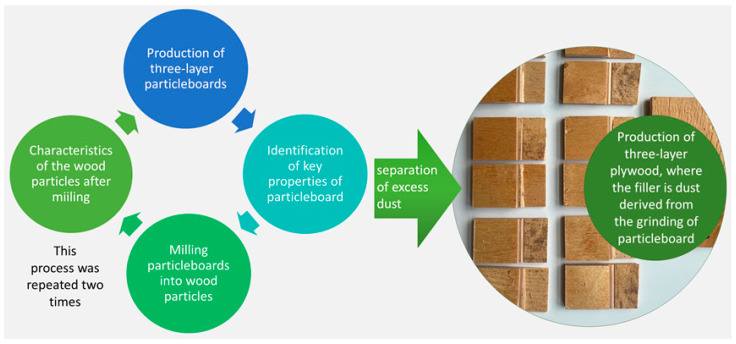
The formation process and utilization of particulate fractions.

### 2.2. Preparation of Panels

As part of this research, three-layer plywood was composed using the following fillers: fine, native wood powder < 0.1 mm and wood powder, obtained from the 1st and 2nd milling of three-layer laminated particleboard. For each type of filler, plywood was manufactured with different proportions of the respective filler: reference, 1, 5, 10, and 20%. The adhesive mixture was prepared in Parts By Weight (pbw): 100:4:10:5 (resin:hardener:filler:water). The adhesive mixture was applied to the veneers with a brush, and each layer of adhesive amounted to 180 g m^−2^. The veneers were laid alternately and then pressed in a hydraulic heated press for 7 min (AKE, Mariannelund, Sweden), with a pressing temperature of 140 °C and a unit pressing pressure of 1 MPa. After pressing, the samples were air-conditioned at 20 ± 1 °C and 65 ± 2% relative humidity for 7 days for weight stabilization before testing.

### 2.3. Characterization of Panels

The mechanical tests were performed on a computer-controlled universal testing machine (Research and Development Centre for Wood-Based Panels Sp. z o.o., Czarna Woda, Poland). The following examinations were carried out: modulus of elasticity (MOE); modulus of rupture (MOR), conducted in accordance with applicable standards [41]; and bonding quality in a dry state, in accordance with the EN 314-1 standard [42]. Each test was conducted with as many as eight repetitions. Using the testing procedure specified in the standard for particleboards and fiberboards, swelling (in thickness) after immersion in water was determined and an analysis was conducted for all variants [43]. Additionally, the water absorption test was conducted. A density profile was also obtained for all the variants (three repetitions) using a Grecon DAX 5000 device (Fagus-GreCon Greten GmbH and Co. KG, ALfeld/Hannover, Germany), based on X-ray techniques, with a sampling step of 0.02 mm and a measuring speed of 0.1 mm s^−1^.

### 2.4. Statistical Analysis

Analysis of variance (ANOVA) and *t*-test calculations were chosen to examine (*α* = 0.05) significant differences between the factors and levels, where appropriate, using the IBM SPSS statistics base (IBM, SPSS 20, Armonk, NY, USA).

## 3. Results and Discussion

### 3.1. The Density Profile

Figure 2 shows the density profiles of samples with the native filler compared to the reference variant. The veneers demonstrated a density of about 600–700 kg m^−3^, depending on the sample. The bonding line exhibited a bonding density slightly above 1200 kg m^−3^ for the reference sample. Regarding the native samples, the greatest bonding line density was observed for the 5% and 20% variants, reaching a maximum bonding line density of approximately 1100 kg m^−3^. Differences in the bonding line densities within the same variant were also observed, which may be explained by excessively high viscosity during the glue application, resulting in uneven bonding.

Figure 3 presents the density profiles of samples utilizing the dust derived from the 1st grinding of the particleboard as the filler, compared to the reference panel. For the 5%, 10%, and 20% variants, the maximum bonding line density remained in assemblance and amounted to approximately 1000 kg m^−3^, whereas a lower density was observed for the 1% variant, equaling around 900 kg m^−3^. As the level of filler contamination increased, the quality of bonding deteriorated, resulting in reduced strength parameters. It also contributed to the disintegration of most samples during the water absorption test. Therefore, future research into these variants should consider modifying the process and incorporating a coupling agent that enhances bonding quality, similar to the approach used to produce five-layer panels using both primary and recycled high-density polyethylene films with various loadings (0, 3, 6, 8, and 11% by weight) of maleic anhydride-grafted polyethylene (PE-g-MA). This research confirmed the positive effectiveness of such an approach [44].

Figure 4 illustrates the density profile of plywood, with the filler consisting of dust derived from the 2nd milling of the boards. In the reference variant, the joint density defaulted to approximately 1200 kg m^−3^. Plywood with a 1% filler exhibited a joint density of approximately 900–1000 kg m^−3^, whereas plywood with 5% and 10% filler obtained values of around 1000–1100 kg m^−3^. In these instances, the glue was absorbed into the wood, as per standard procedures. The highest glue absorption into the wood was observed for the variant with a filler content of 20%. Consequently, the joint density in this case performed lower, ranging from 800–900 kg m^−3^. This phenomenon can be attributed to the increased presence of impurities in the glue within the dust obtained after the 2nd milling. The plywood samples with repeatedly ground filler were thicker. As a result, this particular dust was less absorbent compared to the other fillers, facilitating easier absorption of the glue mixture into the wood. Research has confirmed that every time the wood materials are milled, there is an increase in wood impurities [39] and a reduction in the size of the wood particles [45].

### 3.2. Bonding Quality

Figure 5 displays the results of the shear strength examination. The shear strength value increased with increasing filler content. The highest values were observed for the reference samples, with rye flour serving as the filler, obtaining a shear strength value of 1.83 N mm^−2^ when the filler content equaled 10%. Another variant affecting shear strength is the utilization of dust obtained from the first grinding of three-layer particleboard as a filler in the sample. The highest shear strength values were recorded for the samples with 20% filler content, with a shear strength value reaching 1.68 N mm^−2^. The panels with 5% filler content were the weakest, with a shear strength value of 1.25 N mm^−2^. Regarding shear strength, the weakest values were found for the samples with filler consisting of native dust and dust from the second grinding of particleboard panels, although they were very similar. The lowest values were 0.97 N mm^−2^ for the native filler and 1.08 N mm^−2^ for the second grinding filler, whereas the highest values reached 1.30 N mm^−2^ and 1.39 N mm^−2^, respectively. The difference between the fillers incorporated can occur due to the origin of the raw material. Flour and wood differ from one another and can absorb moisture differently regarding their structure. To identify the key differences for consideration, a chemical analysis of the raw materials, as in the case of red alder bark and walnut shell, is required [46]. Additionally, some of the fillers used have already been modified and contain adhesive additives, including dust obtained from grinding boards. Other studies conducted on dust derived from wood sanding [47] confirmed that the particle size and its pH remain significant factors in selecting a filler for UF resin-based binders. This research confirmed that the smaller the particle size, the higher the strength results, reaching up to 63% when considering particles smaller than 0.125 mm. Furthermore, the smaller the particle size, the lower the filler consumption. The use of waste lignocellulosic fibers from the fiberboard industry, pulp, and paper mills as a filler in plywood revealed that increasing the proportion of this type of filler had a detrimental effect on the shear strength parameters. Only smaller amounts slightly improved the strength, with approximately 1–3% of this filler, whereas increasing the proportion of filler led to a deterioration of shear strength [48]. The statistically significant differences in shear strength within the same filler were found exclusively for the highest and lowest amounts of native filler content. When analyzing all the tested variants, the reference samples were statistically significantly distinct from all the other variants, excluding 10 and 20 pbw 1st milling fillers and 20 pbw 2nd milling filler.

### 3.3. Modulus of Rupture and Modulus of Elasticity

The chart below (Figure 6) depicts the modulus of rupture (MOR) for the manufactured plywood samples. The highest MOR value was found for the reference samples, resulting in a value of 147 N mm^−2^. The other variants performed less favorably compared to the reference sample. For the native sample and the sample with filler obtained after the second grinding of particleboard, the highest values were recorded for plywood with a 20% filler content, equaling 87.2 N mm^−2^ and 68.7 N mm^−2^, respectively. For plywood that contained the filler obtained after the first grinding of particleboard, the highest value was also achieved for the 20% variant, at 69.5 N mm^−2^. The chart also illustrates that increasing the filler content in this case has a limited impact, as even with a 10% filler, the MOR result was similar, at 68.9 N mm^−2^. For lower filler shares, the modulus of rupture (MOR) was lower; however, in this instance it was the least dynamic compared to the other fillers. The most significant differences in the applied filler amounts were observed for the native filler. In comparing the obtained MOR results with data from the literature, considering 10–15% as the optimal filler content, we concluded that the presented results remain similar, since increasing the amount of filler does not bring significant changes [49]. Statistically significant differences in the MOR values within the same filler were found for the lowest native filler content when compared to 10 and 20 pbw, as well as for 20 pbw after 2nd milling when compared to the remaining variants. When analyzing all the tested variants, the reference samples were statistically significantly distinct from all the others. Similarly, the samples containing 20 pbw native filler were statistically significantly distinct from the remaining variants.

The modulus of elasticity (MOE) values of the produced samples are displayed in Figure 7. Again, the greatest MOE values were observed for the reference variant at 14,734 N mm^−2^. The second highest MOE result was recorded for the samples with filler obtained after 2nd grinding, where the filler content was 20% (12,165 N mm^−2^). For this variant, transitions in the filler content had the most significant impact in comparison to the other variants. A similar trend was maintained for the native variant; however, the final MOE parameter for 20% filler content was lower than that of filler obtained after 2nd grinding (11,445 N mm^−2^). Analyzing the MOE results obtained for plywood with filler obtained after 1st grinding, it can be concluded that the filler content did not significantly affect the outcome of this study. The highest MOE value was recorded for a filler content of 20%, reaching 13,166 N mm^−2^. This may also have been responsible for the diminished strength parameters [50]. Evidence from previous research suggests that an excessive proportion of filler can lead to excessive glue viscosity, making its application more challenging [51]. Statistically significant differences in the MOE values within the same filler samples were found for the lowest native filler content and filler obtained after 2nd milling (10 and 20 pbw). When analyzing all the tested variants, the reference samples were statistically significantly distinct from all the others, excluding 20 pbw 1st milling filler and 20 pbw 2nd milling filler.

### 3.4. The Thickness Swelling and Water Absorption Tests

Figure 8 illustrates the results of thickness swelling of the samples manufactured with the investigated alternative filler content of 20 pbw. The first numbers (0/20, 1/20, and 2/20) indicate the type of filler used, respectively, native (0), after 1st milling (“1”), and after second milling (“2”). In the initial phase, the specimens exhibited the most dynamic swelling. After two hours, a large proportion of the test specimens disintegrated. It should be noted that regular urea-formaldehyde resin (non-water resistant) was used in this research. The greatest swelling was observed for the samples using dust obtained from the first milling as a filler. Intermediate values were obtained from the samples with native filler and after the second washing. The minimum swelling per thickness was detected for samples containing rye flour as a filler.

The water absorption results are shown in Figure 9. The highest water absorption was recorded for the sample using native filler. After 24 h of soaking, water absorption was similar for the reference sample and the sample using the 1st milled filler. Despite the initially rapid absorption dynamics, the sample using filler dust obtained after the 2nd grinding eventually showed the lowest water absorption value after 24 h. One area necessitating enhancement in the conducted research is the adhesion quality, notably evident during the water absorption tests. A great portion of the samples exhibited disintegration under these conditions. Consequently, given that a substantial rise in moisture content is adverse for these wood composites, even slight fluctuations in moisture can lead to a decline in the strength parameters [52].

## 4. Conclusions

This series of studies is a continuation of research conducted under fully controlled conditions and still remains a fairly new approach to recycling. Previous studies have shown that the regrinding of raw materials leads to the formation of numerous amounts of dust fractions. Therefore, this paper was devoted to one of the potential methods for managing dust in lignocellulosic layered composite technology. The dust obtained was distinct in its composition in terms of the number of chemical additives, with the largest proportion of such fractions found in dust obtained from the second milling of particleboard. An important aspect to consider when analyzing the results is the origin of the raw material, since commercially used flour has a structure different to wood and remains clean and contaminated. However, utilizing upcycled wood dust can enhance the value of waste materials and is a promising outcome in terms of the principles of a circular (closed-loop) economy, waste upcycling, and carbon capture and storage (CCS) policies.

According to the results achieved, the highest MOE values were observed for the reference samples. Conversely, for samples featuring filler material from the initial milling process, there were no substantial differences in filler content. Notably, in samples where the filler was sourced from the secondary milling of particleboard, the MOE values exhibited an upward trend in conjunction with increasing filler content. Meanwhile, the modulus of rupture (MOR) decreased with an elevated degree of filler milling from 73.1 N mm^−2^ for native filler, through to 68.9 N mm^−2^ for filler after 1st milling, and to 54.5 N mm^−2^ for filler after 2nd milling (with 10 pbw of filler used as a reference), though it increased slightly as the filler content increased. With regard to shear strength testing, the most favorable outcomes were achieved in the samples incorporating filler material from the initial milling of particleboard. The thickness swelling of the plywood reached its peak in variants utilizing filler material from both the initial and secondary milling of particleboards (20.1% and 16.6% after 24 h of soaking for samples with 10 pbw filler after 1st and 2nd milling, respectively, compared to 13.0% for the reference samples). In contrast, water absorption testing exhibited a more pronounced response in the newly introduced variants. Although, samples incorporating filler from the initial and secondary milling processes eventually yielded results akin to the reference samples, with naturally occurring dust displaying higher water absorption values. In terms of the density profiles, the highest density values (about 1224 kg m^−3^) were observed in samples utilizing rye flour as the filler material. A similar density profile was observed in samples with five parts of wood flour used as a filler; however, the density of the bonding line was slightly lower in these cases (1130 kg m^−3^).

The use of fillers in this form allows the substitution of conventionally utilized grain flour, contributing a significant part of the food chain. Nevertheless, the parameters obtained are not as high as expected. Based on the conducted tests, we conclude that it would be advisable to chemically modify the recovered dust to accelerate its moisture absorption capabilities. Also, the hygienic aspects, like volatile organic compounds (VOC) or formaldehyde emissions of the composites produced using recycled wood dust, will be the subject of further research in this field.

## Figures and Tables

**Figure 2 materials-16-07352-f002:**
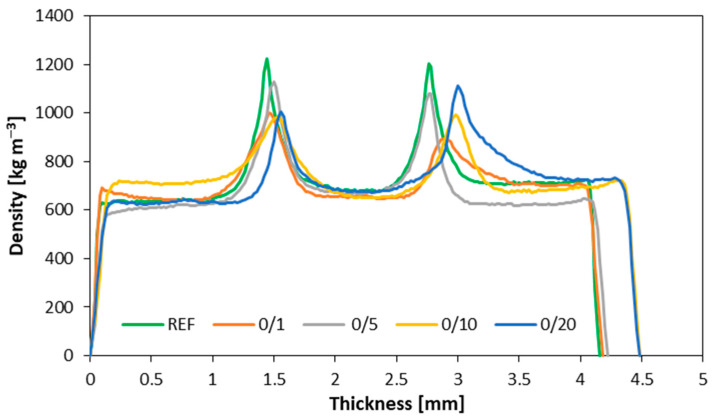
The density profiles for reference and native variants.

**Figure 3 materials-16-07352-f003:**
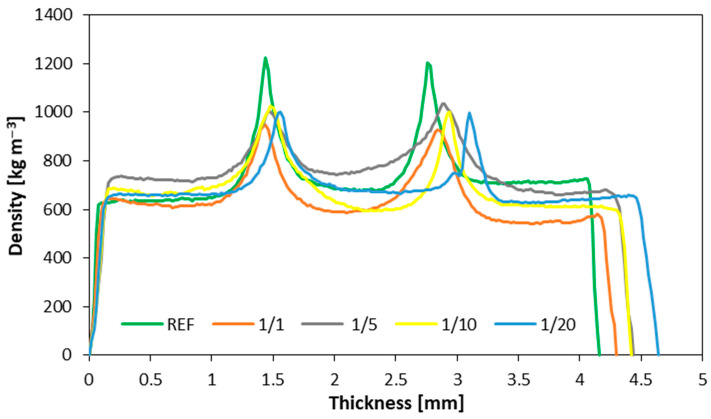
The density profiles for dust after first milling.

**Figure 4 materials-16-07352-f004:**
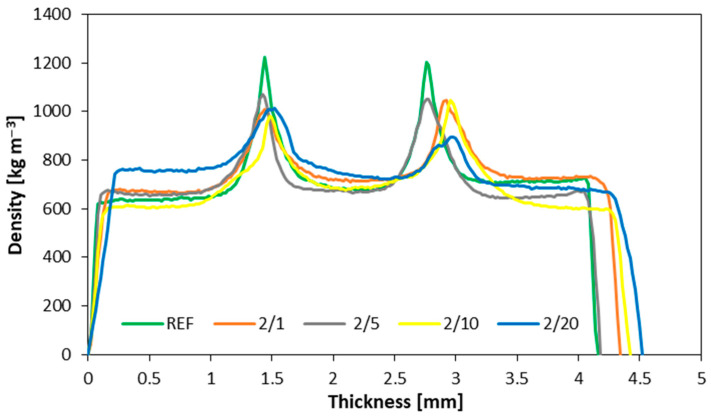
The density profiles for dust after second milling.

**Figure 5 materials-16-07352-f005:**
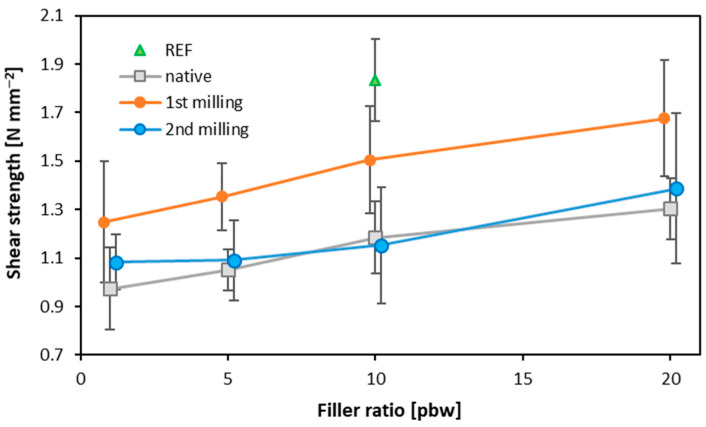
The shear strength of plywood samples of different filler content.

**Figure 6 materials-16-07352-f006:**
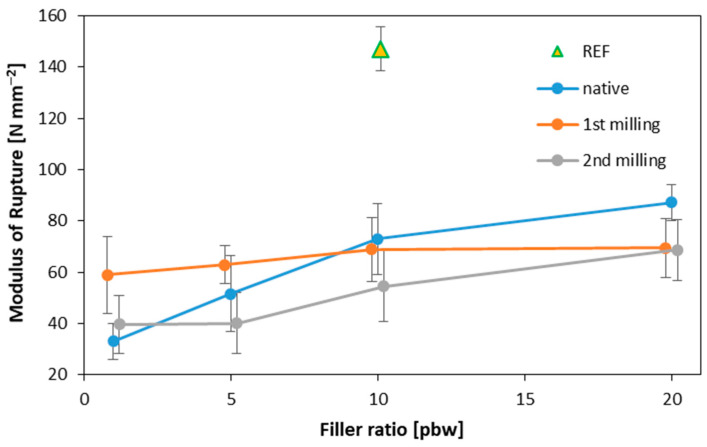
The modulus of rupture of plywood samples of different filler content.

**Figure 7 materials-16-07352-f007:**
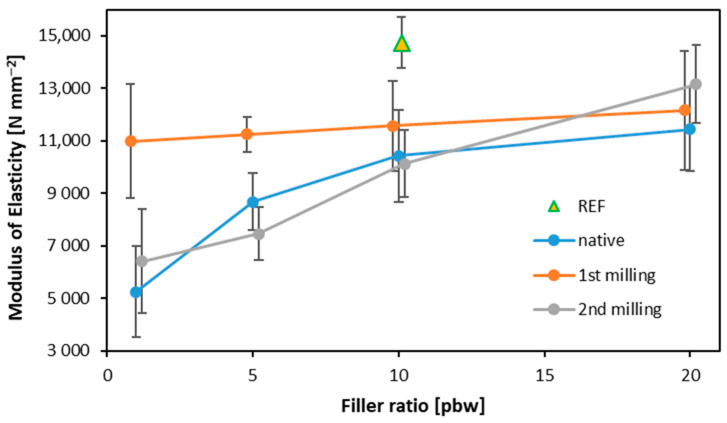
The modulus of elasticity of plywood samples of different filler content.

**Figure 8 materials-16-07352-f008:**
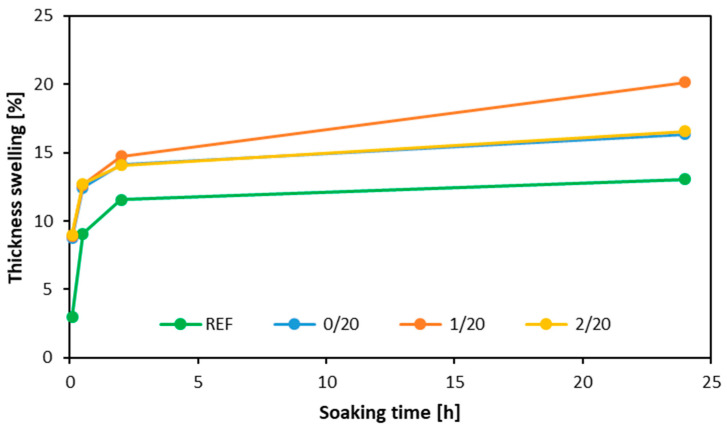
Thickness swelling of the tested plywood with different fillers.

**Figure 9 materials-16-07352-f009:**
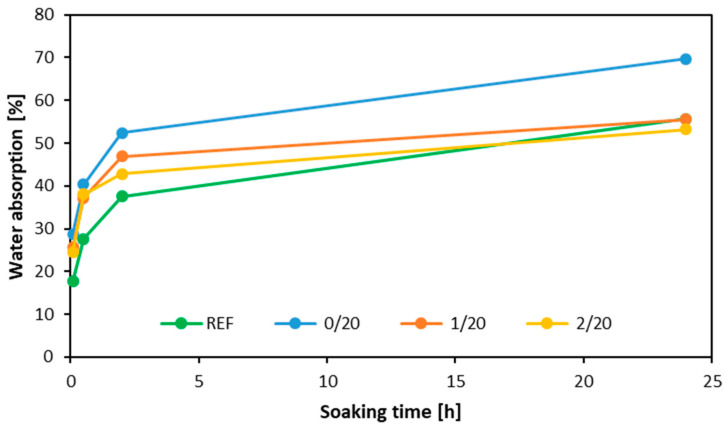
Water absorption of the tested plywood with different fillers.

## Data Availability

The data presented in this study are available on request from the corresponding author.

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
