# Peer review of "Upcycling of Wood Dust from Particleboard Recycling as a Filler in Lignocellulosic Layered Composite Technology"

_materials, 2023, doi:10.3390/ma16237352_

Round 1
Reviewer 1 Report
Comments and Suggestions for Authors
The manuscript is focused on investigation and evaluation of the feasibility of manufacturing 3-layer plywood fabricated with addition of wood dust, obtained from particleboard recycling. In this respect, the manuscript is within the scope of the Materials journal. In general, the manuscript is well-developed, structured, and informative, however, it needs some revisions before acceptance for publication.
The title (lines 2-3) and the keywords (line 36) correspond to the aims and objectives of the manuscript.
The abstract (lines 8-35) is informative, and contains the main findings of the research. However, it is rather long and should be shortened. Moreover, I’d recommend adding more specific information (results) obtained from the experimental work, e.g. the respective MOR and MOE values, as well as some data related to the physical properties (density, thickness swelling and water absorption).
Lines 8-9: the first sentence of the abstract is not necessary and can be omitted.
Line 46: “e.g.” is not needed, please revise.
Line 49: “formaldehyde glue”: it is more correct to say “formaldehyde-based thermosetting resins”, please revise.
Lines 56-57: “formaldehyde-reducing properties”: it would be more correct to state, e.g., “decreased free formaldehyde emission”.
Lines 59-60: the statement “Another raw material that allows formaldehyde emissions is a powder derived from modified ground pine needles.” is not clear and should be revised.
Line 65: “Thymus Plants”: plants should not be capitalised, please revise.
Lines 66-69: this statement belongs to the very beginning of the manuscript, please revise.
Line 71: there is full stop after “cost”.
Line 79: “we distinguish”: please revise.
Line 94: “UF resin”: please add the full term, i.e., urea-formaldehyde resin, then the common abbreviation “UF”.
Lines 95-96: “…this rather interesting piece of information can serve as inspiration for further processing of used dust in subsequent research.”: this does not bring any information to the reader and can be deleted/revised.
Lines 101-102: who are these researchers? Please add relevant reference(s).
Line 108: what do you mean by “recycled adhesive”?
Line 119: please add the standard in the references of your work.
Lines 132-134: please specify more clearly the aim of your research work.
Line 137: I believe the place of this Figure is in the Materials and Methods section, and not in the Introduction.
Overall, the Introduction part is informative and provides some relevant information on the topic of the research, based on previously published studies. However, it should be further elaborated based on the above comments.
Line 145: could you provide some additional data on the UF resin used, besides the solids content?
Line 147: this statement should be revised; the verb is missing.
Line 166: relative humidity, please revise.
Line 167: What was the duration of samples’ conditioning?
Overall, the Materials and Methods section is well prepared and provides relevant information on the materials, methods, and equipment used to conduct the experimental work.
Line 186: “attached chart”: what do you mean by that?
Line 205: UF resin, not “glue”, please revise.
Line 263: it should be “thickness swelling”, please revise.
Line 271, Figure 5: “swelling” should not be capitalized, please revise.
Line 283, Figure 6: “absorption” should not be capitalized, please revise.
The same comments apply also for the captions of figures 7-9.
In general, the results of the study are informative and properly discussed with relevant research works in the field.
The Conclusion part should be carefully check and revised, it should include a summary of the main findings of the research. The information, included in lines 327-338 should be either revised or deleted, now it is too general and not relevant.
The references cited are appropriate and correspond to the topic of the manuscript.
Comments on the Quality of English LanguageModerate editing of English language and style used are required.
Author Response
Dear Reviewer,
in the attachment, please find our response to the issues raised in the review.
With best regards
Grzegorz Kowaluk

Reviewer 2 Report
Comments and Suggestions for Authors
The research presented here is focused on the selected properties of 3- layer plywood, produced with the utilization of dust from the milling of three-layer particleboard as a filler in the bonding mass. Four types of filler were used in the study: commercial rye flour, wood dust naturally occurring in the composition of particles used industrially for particleboard production, wood dust from the first batch of shredded particleboard, and dust from the second round of milled particleboard. The research confirms the possibility of using the aforementioned dust as an alternative to conventional fillers in plywood technology. It also raises the question of how to effectively remove glue residues from wood-based composite dust, which would enhance their absorption properties. However, there are several questions not mentioned or clearly clarified by the authors, so the manuscript should be revised before publication.
1) There are problems with the formatting of the paragraphs in this article.
2) The introduction to this article is too long and needs to be streamlined.
3) The overall language accuracy and logical writing of this paper is poor.
4) The process of formation and utilisation of particle components is described only in words, are relevant physical or 3D pictures added?
5) The text lacks specific information on the four added ingredients, such as particle size and density.
6) The text lacks microscopic characterization and chemical analyses of raw materials as well as specimens, and it is suggested to add them.
7) The error bars for the performance of each specimen are large, and it is recommended that multiple tests be conducted to reduce the error bars.
8) REF in the text only has the individual properties for the 10 per cent additive, and it is recommended that the properties for the other additives be supplemented.
9) Only the sectional density of the specimen is explored in the paper, and it is suggested that specific density values be added.
10) It is recommended that the profile density analysis be placed in front of the mechanical property analysis, which is more logical to the writing of the article.
11) The individual figure notes in Figures 5 and 6 are not explained in the text and it is suggested that they be supplemented and improved.
Comments on the Quality of English Languagetotally good
Author Response

(The authors gave the same response as above.)

Reviewer 3 Report
Comments and Suggestions for Authors
This paper presents an interesting study on lignocellulosic layered composites technology using dusts from the milling of three-layer particleboard as a filler in the bonding mass. However, there are a few concerns that need to be clarified with some further suggestions and comments.
1, Title is not very clear to represent the topic studied.
2, The abstract needs to rewrite as it is not clear and summarised.
3, The introduction needs to improve. Line 66-69 seems it should be presented at the beginning. In some parts, references seem missing such as Line 100-102.
4, Line 147-149, as fillers used, the physical properties of the fillers have not been given such as particle size ranges. Between the 1st and 2nd milling, what difference does the dust have? As the dusts came from the milling of 3-layer laminated particleboards, what may the chemical components of the dusts have?
5, Line 168, The methods for characterisation analysis of the test samples are not clear. What have the tests been done, tensile or shear tests? What is the sample size?
6, Line 181, where does the analysis of variance present?
7, The conclusion needs to improve as it is not clear on what it has been learned from the study, If the dust fillers are not as good as the conventional fillers as shown in the conclusion, why do we need to use the dust fillers? This should be quantified in the introduction and concluded here.
Other suggestions are as follows:
1, Line 75: ‘biomass combustion fly ash’ sounds quite different to the other stuffs mentioned here, chestnut flour, rice starch, and soy flour.
2, Line 188: tensile strength or shear strength?
3, Line 198: what is the ‘fundamental difference’? Structure difference between the raw materials was identified as a key factor, but there is not evidence shown here.
4, Line 205: ‘This research confirmed that the smaller the particle size, the higher the strength results,’ shows the importance of the particle size.
5, Line 271, Fig. 5: the meaning of the legends is not clear. What do the 0/20, 1/20 and 2/20 mean?
6, Line 284: How does the density profile of the samples measure? The method has not been given.
7, Line 327-330: these should be concluded in the introduction rather than in paper conclusion. It should conclude that what have been studied in the paper directly.
Comments on the Quality of English LanguageThe English needs to improve significantly.
Author Response

(The authors gave the same response as above.)

Round 2
Reviewer 1 Report
Comments and Suggestions for Authors
Dear authors, thank you for addressing all my comments/remarks on your work. I believe the revised manuscript can be accepted for publication in its current form.
Comments on the Quality of English LanguageEnglish language and style are fine, only some minor issues should be addressed.
Author Response
Dear Reviewer,
thank you for accepting our efforts to improve the manuscript. We have transferred it to additional language polishing.
With best regards,
Grzegorz Kowaluk
Reviewer 3 Report
Comments and Suggestions for Authors
The paper has been improved accordingly. Except some small errors in the paper, it seems acceptable for publication.
The errors I mentioned is regarding the English language usage. Some sentences are a bit hard to read and difficult to understand. For example, Line 10, ‘Four types of filler were considered …’ should be ‘Four types of fillers were considered …’. The abstract seems a bit long and unfocused, which has not been improved. A long abstract is never recommended.
Line 134, ‘To reach this aim, the fine fraction of recovered wood material at different content has been applied …’ is not clear and professional.
There are quite other bits I do not like it. Technically, the paper seems fine but in presentation the language seems a bit poor.
Comments on the Quality of English Language
Extra editing of English language is recommended.
Author Response
Dear Reviewer,
thank you for accepting our efforts to improve the manuscript. We have corrected the language errors you indicated, and we have transferred the manuscript to additional language polishing.
Also, we shorten the abstract.
With best regards,
Grzegorz Kowaluk